# Characteristics of CD5-positive diffuse large B-cell lymphoma among Koreans: High incidence of BCL2 and MYC double-expressors

Hee Young Na[1,2], Ji-Young Choe[2,3], Sun Ah Shin[2,4], Hyun-Jung Kim[5], Jae Ho Han[6], Hee Kyung Kim[7], So Hee Oh[8], Ji Eun Kim[2,9]*

1 Department of Pathology, Seoul National University Bundang Hospital, Seongnam, Republic of Korea,
2 Department of Pathology, Seoul National University College of Medicine, Seoul, Republic of Korea,
3 Department of Pathology, Hallym University Sacred Heart Hospital, Anyang, Gyeonggi, Republic of Korea,
4 Department of Pathology, Seoul National University Hospital, Seoul, Republic of Korea, 5 Department of Pathology, Inje University Sanggye Paik Hospital, Seoul, Republic of Korea, 6 Department of Pathology, Ajou University School of Medicine, Suwon, Geonggi, Republic of Korea, 7 Department of Pathology, Soonchunhyang University Bucheon Hospital, Bucheon, Republic of Korea, 8 Department of Biostatistics, Seoul National University Seoul Metropolitan Government Boramae Medical Center, Seoul, Republic of Korea, 9 Department of Pathology, Seoul National University Seoul Metropolitan Government Boramae Medical Center, Seoul, Republic of Korea

* npol181@snu.ac.kr

**Data Availability Statement:** All relevant data are within the paper and its Supporting Information files.

## Abstract

Aberrant expression of CD5 has been reported in 5–10% of diffuse large B-cell lymphomas (DLBCLs). CD5+ DLBCL had been recognized as an aggressive immunophenotypic subgroup of DLBCL in the 2008 WHO classification of haematolymphoid neoplasm; however, it was eliminated from the list of subgroups of DLBCLs in the revised 2016 classification. Nevertheless, there is much controversy regarding the clinical significance of CD5 expression, and many researchers still assert that this subgroup exhibits an extremely unfavorable prognosis with frequent treatment failure. We retrospectively investigated 405 DLBCLs recruited from three university hospitals in Korea from 1997 to 2013. The clinical profile, immunophenotype, and chromosomal structural alterations of the BCL2 and MYC genes were compared according to CD5 expression. A total of 29 cases of *de novo* CD5+ DLBCL were identified out of 405 in our series (7.4%). Clinicopathologic correlation was performed in all 29 CD5+ DLBCLs and 166 CD5- DLBCLs which were eligible for full clinical review and further pathologic examination. Compared with CD5- counterparts, CD5+ DLBCLs showed female preponderance, frequent bone marrow involvement, higher lactate dehydrogenase level, advanced Ann Arbor stages and poorer prognosis (all p<0.05). Pathologically, the expression of CD5 positively correlated with that of BCL2, MYC and Ki-67 (all p<0.05). Coexpression of BCL2 and MYC, which is referred to as a double-expressor, was relatively more common in CD5+ DLBCL, whereas translocation or amplification of these genes was very rare. in conclusion, the expression of CD5 is an independent poor prognostic factor of DLBCLs, and this subgroup displays unique clinicopathologic features. Although the exact

**Funding:** This study was supported by the Grant of National Research Foundation of Korea (2017R1A2B4005052). The funders had no role in study design, data collection and analysis, decision to publish, or preparation of the manuscript.

**Competing interests:** The authors have declared that no competing interests exist.

mechanism remains uncertain, consistent activation of BCL2 and MYC by alternative pathways other than chromosomal translocation may contribute to the pathogenesis.

## Introduction

Pathologic diagnosis of malignant lymphoma is based on the application of immunohistochemistry (IHC) using lineage-specific surface markers such as CD3 or CD20. However, aberrant expression of some T-cell markers, of which the most representative is CD5, has been well documented in a subset of B-cell neoplasms [1–3]. In fact, CD5 is an important diagnostic marker of chronic lymphocytic leukemia/small lymphocytic lymphoma (CLL/SLL) and mantle cell lymphoma [4].

The expression of CD5 in diffuse large B-cell lymphoma (DLBCL) can be observed in Richter transformation of CLL but can also be found in *de novo* DLBCLs. Since it was first recognized in 1995 [2], many *de novo* CD5+ DLBCL cases have been documented, and the overall incidence comprises 5–10% of all DLBCLs [1, 5, 6]. The CD5+ DLBCL had been introduced as an immunophenotypic subgroup of DLBCL in the 2008 WHO classification of haematolymphoid neoplasms, however, the revised 2016 version has omitted designation of the CD5+ subtype. Nevertheless, accumulating evidences suggest that CD5+ DLBCL is a distinctive subgroup which typically presents aggressive clinical features and adverse outcomes [1, 4, 7–11]. Previous studies have confirmed that the prognosis of CD5+ DLBCL is still poor regardless of Rituximab based chemotherapy [1, 9, 10], and even with the salvage stem cell transplantation [12]. To achieve optimal therapeutic responses, better understanding of pathogenic mechanisms and risk stratification are crucial.

To date, most large-scale studies of *de novo* CD5+ DLBCL have been performed in Japan, and there are only few reports from other Asian countries or Western areas [1, 8, 9, 12–15]. We performed a retrospective study to review detailed characteristics of CD5+ DLBCL among Korean patients, particularly focusing on the relationship to other constitutional prognostic factors, such as cell of origin by IHC, and BCL2 and MYC status.

## Materials and methods

### Case selection and analysis of the clinical characteristics

Cases diagnosed as DLBCL, not otherwise specified (NOS), were retrieved from three university hospitals (Seoul National University Hospital, Seoul National University Bundang Hospital and Seoul National University Boramae Hospital) in Korea from January 1996 to January 2016. The diagnosis was confirmed by two experienced hematopathologists (HYN and JEK), based on the 2017 WHO classification of Tumours of Haematopoietic and Lymphoid Tissues [4]. Clinical profiles and follow-up data were obtained from electronic medical records.

### Immunohistochemistry (IHC) and Epstein-Barr virus (EBV) detection

To determine the CD5+ subgroup, all DLBCL cases were reexamined and assessed by IHC. IHC was performed using 4 μm sections of paraffin-embedded tissue blocks using the following antibodies BCL2 (M0887, mouse, monoclonal, 1:100; Dako, Carpinteria, CA, USA), BCL6 (LN22, mouse, monoclonal, 1:100; Novocastra, Newcastle, UK), CD3 (M7254, mouse, monoclonal, 1:100; Dako), CD5 (M3641, mouse, monoclonal, 1:100; Dako), CD10 (PA0270, mouse, monoclonal, 1:100; Novocastra), CD20 (M0755, mouse, monoclonal, 1:400; Dako), IRF4/

MUM1 (M7259, mouse, monoclonal, 1:100; Dako), Ki-67 (M7240, mouse, monoclonal, 1:100; Dako) and MYC (Y69, rabbit, monoclonal, 1:100; Epitomics, Burlingame, CA, USA). The IHC for antibodies were performed. First, sections were treated with Target Retrieval Solution (Dako, Glostrup, Denmark) at 115˚C for 15 min after inhibiting endogenous peroxidase activity for 30 min with 3% hydrogen peroxidase in methanol for antigen retrieval.Then, immune complexes were detected with the Envision Detection System (Dako) after overnight incubation. Finally, hematoxylin counterstaining was done.

For CD5 IHC, we used the cutoff of more than 50% tumor cells showing immunoreactivity for CD5 of any intensity as positive. Tumor cells with more than 30% staining were considered positive for BCL6, CD10 and IRF4/MUM1 [16]. For BCL2 and MYC, staining of more than 50%, and 40% was used as cutoff, according to the criteria suggested by Johnson et al [17]. EBV in situ hybridization (ISH) was performed using an EBV-encoded RNA (EBER) probe (INFORM EBER Probe; Ventana Medical Systems, Tucson, AZ, USA). The evaluation of IHC and EBV ISH were perfomed by the same two experienced hematopathologists (HYN and JEK), and discordant cases were discussed for consensus.

## Detection of IgH/MYC translocation, MYC amplification and IgH/BCL2 translocation

Fluorescence in situ hybridization (FISH) was performed on paraffin embedded tissue block sections according to the manufacturer's protocol. A Vysis LSI MYC dual-color, break-apart rearrangement probe (Abbott Molecular, Abbott Park, IL, USA) was used to detect MYC translocation, and a Vysis IgH/MYC/CEP 8 Tri-color DF probe (Abbott) was used for amplification. At least 100 cells from each case were assessed for split signals to identify MYC translocation and gene copy number alteration. FISH using a Vysis LSI BCL2 dual-color break-apart rearrangement probe (Abbott Molecular, Abbott Park, IL) was performed to identify BCL2 translocation. The result was considered positive for rearrangement and amplification when >20% of nuclei showed a break-apart signal or extra copies [18].

## Statistical analysis

For the association analysis, the Mann–Whitney U-test, Fisher's exact test or Pearson's chi-square test was performed. Spearman's ρ was used to assess correlations between variables. Overall survival (OS) was measured from the date of diagnosis to the date of death. Progression-free survival (PFS) was estimated from the date of diagnosis to the date of disease progression, including relapse and death. Univariable Kaplan–Meier survival analysis with log-rank tests was conducted to compare the outcome based on the status of various parameters. Multivariable survival analysis using the Cox proportional hazards model was performed to identify independent prognostic markers. A two-tailed P-value of ≤0.05 was considered statistically significant. All data were analyzed with SPSS software, version 22.0 (SPSS Inc., IBM, Armonk, NY, USA).

## Ethics

Ethical approval was obtained by the regional ethics committee in Seoul National University Boramae Hospital, Korea according to the declaration of Helsinki (2014/020, 2014/020/1, and 2014/233) (IRB No. 10-2018-19). Informed consent from participants was not required according to the ethics committee. All data were fully anonymized before accessed.

## Results

A total of 405 cases of DLBCL were investigated for CD5 expression on tumor cells, and 30 cases of CD5+ DLBCLs, consisting of one case transformed from CLL and 29 *de novo* cases (29 of 405, 7.2%), were identified. To compare the clinicopathologic findings between the CD5 + and CD5- groups, a total of 166 CD5-DLBCL, which had no evidence of transformation from low grade lymphoma, were selected, of which detailed clinical data and sufficient tissue for further pathologic studies were available. Clinicopathologic profiles of *de novo* CD5 + DLBCL (N = 29) and CD5-DLBCL patients are summarized in Table 1. Most CD5- DLBCL patients (21/29, 72.4%) were treated with CHOP (cyclophosphamide, doxorubicin, vincristine, and prednisone)-based chemotherapy with or without rituximab. Of these, 3 received etoposide in addition to the above combination (R-EPOCH). The remaining 4 patients were

**Table 1. Comparison of clinicopathologic features of CD5+ and CD5- DLBCL patients.**

| Clinicopathologic Feature | CD5+ DLBCL (n = 29) | CD5- DLBCL (n = 166) | P-value |
|---|---|---|---|
| Age over 65 yr | 15 (51.7%) | 59 (35.5%) | 0.098 |
| Sex (Male: female) | 9:20 | 103:63 | 0.002* |
| Initial diagnosis in lymph node | 16 (55.2%) | 60 (36.1%) | 0.039* |
| BM involvement | 11 (37.9%) | 20 (12.1%) | 0.010* |
| Ann Arbor Stage (I-II:III-IV) | 10:19 | 97:69 | 0.030* |
| ECOG PS (0–1:2–5) | 18:11 | 132:34 | 0.060 |
| Elevated LDH | 20 (69.0%) | 78 (47.0%) | 0.028* |
| >1 Extranodal sites | 13 (44.8%) | 30 (18.1%) | 0.002* |
| B symptoms | 8 (27.6%) | 41 (24.7%) | 0.689 |
| Bulky tumor | 4 (13.8%) | 23 (13.9%) | 0.980 |
| IPI (0–2:3–5) | 11:18 | 113:53 | 0.004* |
| Disease progression | 18 (62.1%) | 83 (50%) | 0.230 |
| CNS relapse | 1 (3.7%) | 8 (4.8%) | 0.745 |
| Treatment | | | 0.065 |
| R-CHOP | 18 (62.1%) | 78 (47.0%) | |
| CHOP | 3 (10.3%) | 51 (30.7%) | |
| Other chemotherapy | 3 (10.3%) | 18 (10.8%) | |
| Other treatment (e.g. Op, RT etc) | 0 (0%) | 9 (5.4%) | |
| No treatment | 4 (13.8%) | 2 (1.2%) | |
| PFS, median (range) | 10 (0–163) | 40 (0–192) | 0.003* |
| OS, median (range) | 13 (1–163) | 44 (0–261) | 0.008* |
| Hans system (GCB:Non-GCB) | 8:19 | 56:110 | 0.570 |
| CD10 | 5 (17.2%) | 26 (15.7%) | 0.830 |
| BCL6 | 14 (48.3%) | 76 (45.8%) | 0.804 |
| MUM1 | 16 (55.2%) | 58 (34.9%) | 0.038* |
| BCL2 | 17 (58.6%) | 50 (30.1%) | 0.003* |
| MYC | 9 (33.3%) | 7 (4.2%) | <0.001* |
| MYC/BCL2 double-expressor | 8 (27.6%) | 5 (3.0%) | <0.001* |
| Ki-67, median (range) (%) | 60 (20–100) | 50 (6.6–88.3) | 0.007* |
| MYC rearrangement | 2 (6.9%) | 6 (3.6%) | 0.614 |

DLBCL, diffuse large B-cell lymphoma; BM, bone marrow; IPI, ECOG, Eastern cooperative group; LDH, lactate dehydrogenase; IPI, international prognostic index; CNS, central nervous system; Op, operation; RT, radiation therapy; PFS, progression-free survival; OS, overall survival; Germinal center B-cell like subgroup.

* P ≤0.05

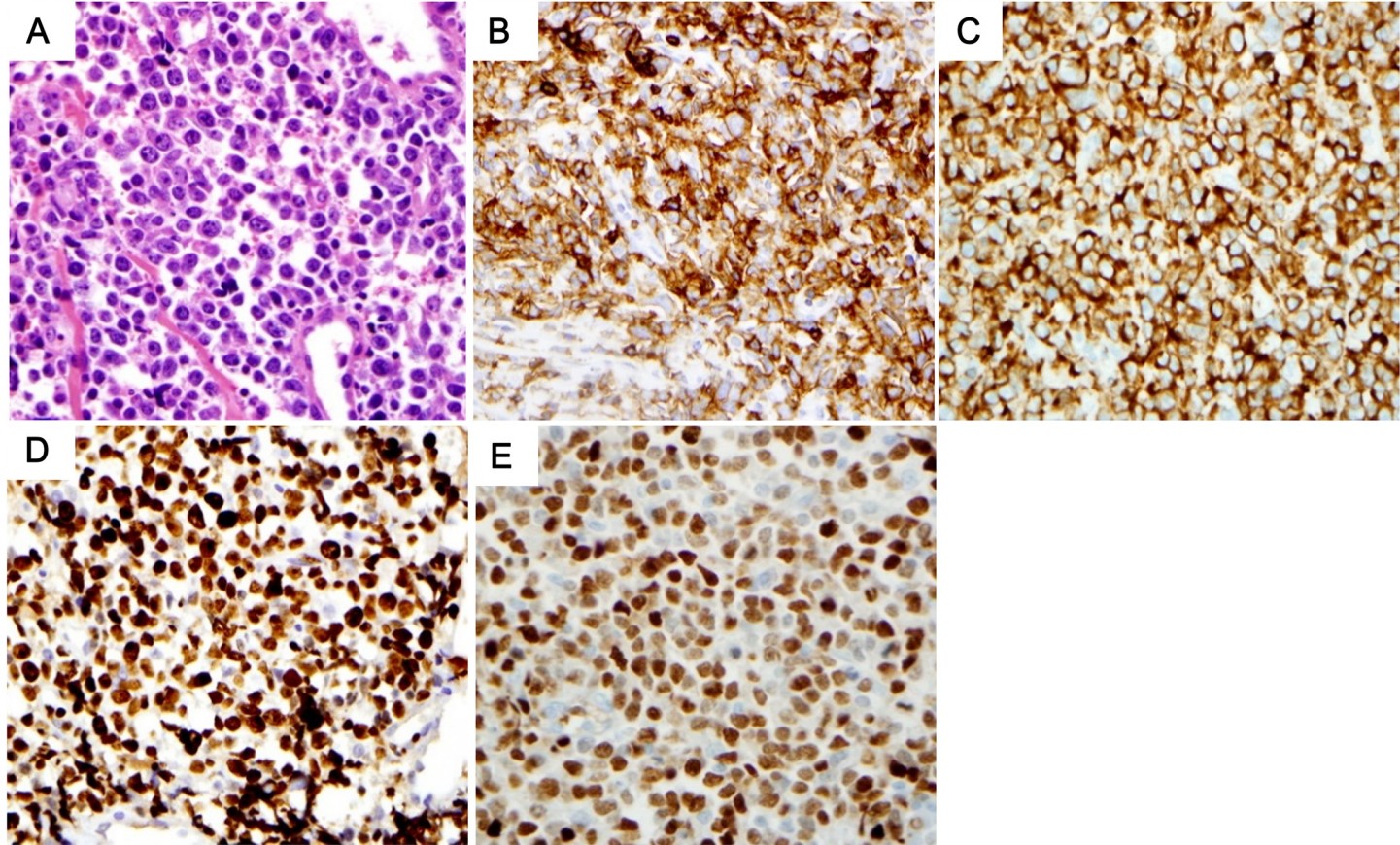

**Fig 1. Pathologic characteristics of CD5+ DLBCL.** Pleomorphic large cells with many apoptotic features were shown in many cases (A). Tumor cells were positive for CD5 (B), BCL2 (C), IRF4/MUM1 (D). Note the diffuse strong pattern in CD5 IHC. Tumor cells also showed high Ki-67 proliferation index (E).

managed by supportive care only. Similarly, 129 (77.7%) CD5-DLBCL patients were treated with a CHOP-based regimen. The median follow-up time of *de novo* CD5+ DLBCL and CD5-DLBCL patients was 13 months (ranging from 1 to 163) and 43 months (ranging from 0 to 261), respectively.

Compared with CD5- DLBCL, the CD5+ group revealed female preponderance (p = 0.002), more than one extranodal site involvement (p = 0.002), and frequent bone marrow involvement (p = 0.01). In addition, elevated lactate dehydrogenase (LDH, p = 0.028) and a higher international prognostic index (p = 0.004) were more common in CD5+ DLBCL.

Both CD5+ and CD5- DLBCL showed non-germinal center B-cell type predominance according to the Hans algorithm [16]. Compared with CD5- cases, CD5+ DLBCL cases showed more frequent IRF4/MUM1 (p = 0.038), BCL2 (p = 0.003), and MYC (p<0.001) expression and a higher Ki-67 proliferation index (p = 0.007). Concurrent expression of BCL2 and MYC, which is known as double-expressor (DE), was significantly more common in CD5 + cases (8/29 vs. 5/166, p<0.001). Of note, only two (2 of 29 CD5+ DLBCLs, and 2 of 9 MYC overexpressed CD5+ DLBCLs) revealed MYC rearrangement, and no amplification was identified by FISH. BCL2 rearrangement or amplification was not found. (Figs 1 and 2).

In the univariable survival analysis, patients with CD5+ DLBCL showed significantly shorter OS (p = 0.025) and PFS (p = 0.033) than those with CD5- tumors (Fig 3), whereas DE lymphomas revealed inferior PFS (p = 0.036) but not OS (p = 0.234) (Fig 4). Additionally,

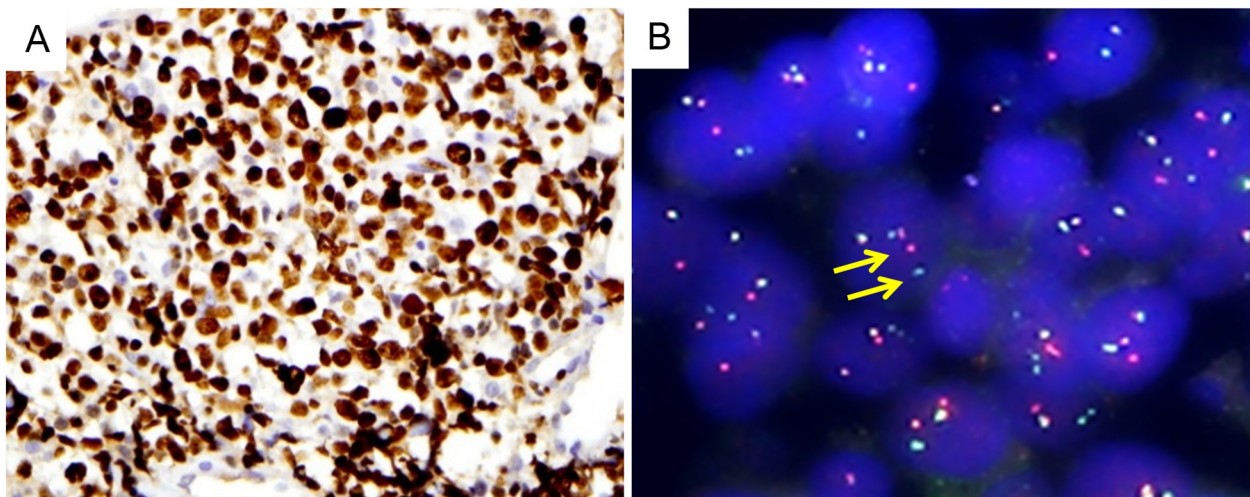

**Fig 2. Representative case showing MYC overexpression and rearrangement.** High expression of MYC protein (A) and presence of MYC gene rearrangement examined by FISH for MYC, break apart probe (B, split signals indicated by arrows).

older age (>65 yrs), higher performance status (PS) by the Eastern Cooperative Oncology Group, elevated serum LDH, presence of B symptoms, more than one extranodal involvement and high Ann Arbor stages were all associated with inferior OS and PFS (all p<0.001) (Tables 2 and 3). In the multivariable Cox regression analysis, older age, higher PS, elevated serum LDH and the presence of B symptoms remained independent prognostic factors for OS and PFS, whereas CD5 positivity and DE did not (Tables 2 and 3).

Among non-GCB type DLBCL, CD5 expression was associated with inferior OS (p = 0.005) and PFS (p = 0.004), while DE was associated only with shorter PFS (p = 0.003) in univariable

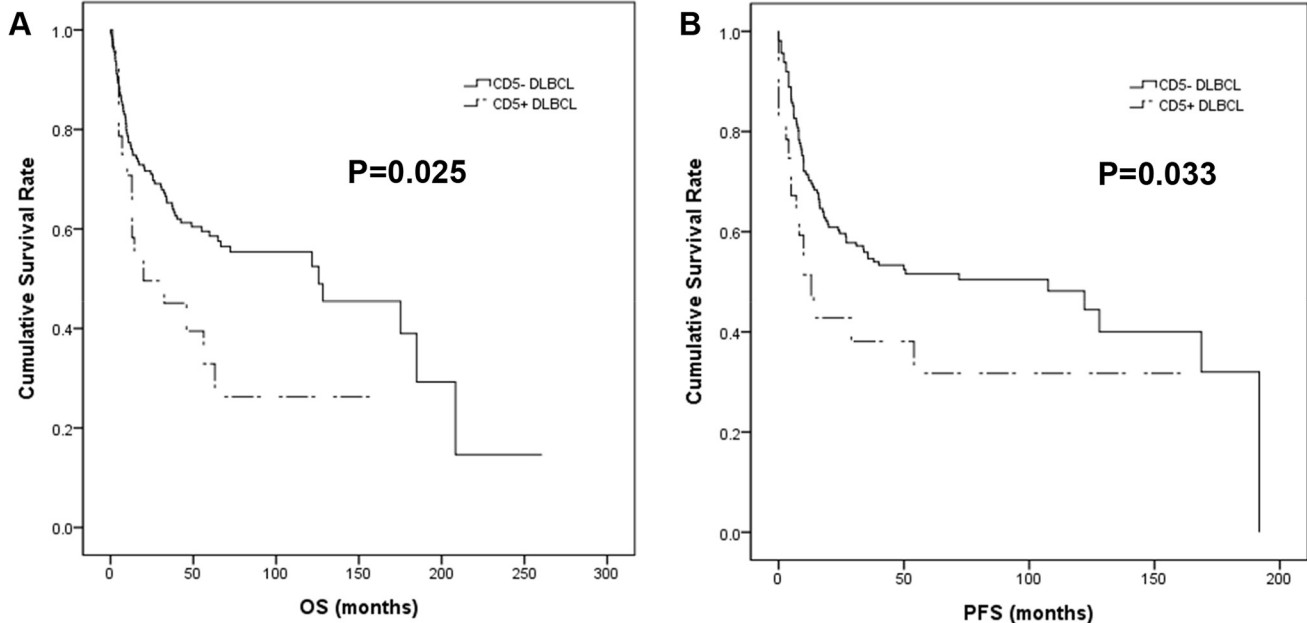

**Fig 3. Kaplan-Meier survival curves of CD5+ and CD5-DLBCL patients.** CD5 positivity was associated with significantly shorter OS (A) and PFS (B).

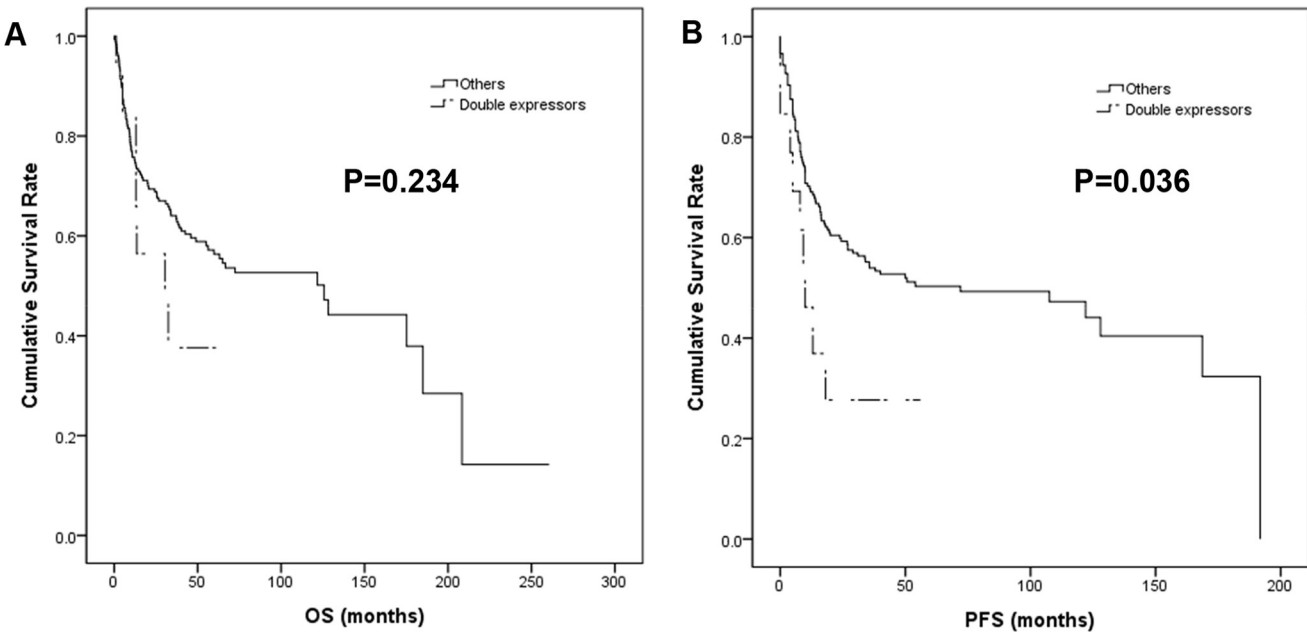

**Fig 4. Kaplan-Meier survival curves of double-expressors compared with others.** DE was associated with inferior PFS (B) but not OS (A).

analysis. In multivariable analysis, CD5 positivity was an independent prognostic factor for poor PFS (p = 0.006), but not OS (p = 0.145) (Tables 2 and 3).

## Discussion

In this study, we demonstrated that CD5 expression was associated with aggressive clinical features and poor survival in DLBCL. Pathologically, the expression of CD5 correlated with BCL2 and MYC positivity, and concurrent expression of these two proteins (DE) was more frequently found in CD5+ DLBCL, whereas chromosomal translocation or amplification was rarely noted. We suggest that CD5+ DLBCL is a distinct immunophenotypic subtype, and its pathogenesis and biologic nature are possibly related to alternative activation of MYC or BCL2 proteins other than gene rearrangement.

The overall incidence of CD5+ DLBCL in our Korean cases accounted for 7.2%, which is similar to previous series from Japan and other Western countries [1, 8, 9, 13–15]. Most of our series were *de novo* cases, because the occurrence of CLL/SLL and mantle cell lymphoma is extremely low in the Korean population, occupying only 2.1% and 1.3% of malignant lymphoma, respectively [19]. Many previous studies identified CD5 expression as an independent prognostic factor regardless of rituximab use [1, 7, 9, 10, 13, 20, 21]. Others also reported that more intensive chemotherapy other than R-CHOP or stem cell transplantation did not overcome its dismal outcome [12, 15, 22]. In the present study, CD5+ DLBCL showed female predominance, higher IPI and Ann Arbor stages, and other aggressive clinical features, which are in line with previous reports [1, 5, 7–9, 13, 14, 20, 21, 23]. These clinical characteristics are also commonly noted in double hit lymphoma (DHL) or DE cases [17, 24, 25]; both have been known as high-risk lymphomas although the clinical behaviors of DE are not as much aggressive as DHLs [26]. In particular, there is considerable pathologic overlap between DE lesions and CD5+ DLBCL regarding cell of origin and complicated mutational events [1, 9, 27–29]. Therefore, we paid special attention to the relationship of CD5 with BCL2 or MYC expression due to the aforementioned resemblance between CD5+ DLBCL and DE lymphomas. We

**Table 2. Univariable and multivariable analysis for OS in DLBCL.**

| Variables | Univariate analysis (OS) | | Multivariate analysis (OS) | |
|---|---|---|---|---|
| | OR (95% CI) | P-value | OR (95% CI) | P-value |
| *All DLBCL* | | | | |
| Age > 65yr | 2.459 (1.622–3.728) | <0.001* | 2.111 (1.245–3.579) | 0.006* |
| Performance status | 3.473 (2.234–5.401) | <0.001* | 2.111 (1.280–3.481) | 0.003* |
| Elevated LDH | 2.362 (1.470–3.797) | <0.001* | 1.915 (1.080–3.395) | 0.026* |
| B symptom | 2.824 (1.828–4.363) | <0.001* | 2.206 (1.327–3.664) | 0.002* |
| > 1 Extranodal sites | 1.796 (1.113–2.900) | 0.017* | 1.037 (0.554–1.941) | 0.909 |
| Stage | 2.260 (1.475–3.464) | <0.001* | 1.308 (0.729–2.346) | 0.369 |
| Double-expressor | 1.465 (0.704–3.049) | 0.234 | 0.583 (0.222–1.531) | 0.273 |
| CD5 positivity | 1.825 (1.071–3.109) | 0.027* | 1.374 (0.723–2.612) | 0.331 |
| *Non-GCB DLBCL* | | | | |
| Age > 65yr | 2.742 (1.642–4.579) | <0.001* | 2.429 (1.260–4.681) | 0.008* |
| Performance status | 3.559 (2.062–6.142) | <0.001* | 1.369 (0.704–2.664) | 0.355 |
| Elevated LDH | 2.687 (1.504–4.801) | 0.001 | 1.784 (0.866 = 3.676) | 0.116 |
| B symptom | 3.026 (1.791–5.115) | <0.001* | 2.456 (1.323–4.558) | 0.004* |
| > 1 Extranodal sites | 1.952 (1.108–3.438) | 0.021* | 0.867 (0.375–2.004) | 0.739 |
| Stage | 2.858 (1.675–4.876) | <0.001* | 2.831 (1.486–5.395) | 0.002* |
| Double-expressor | 2.189 (0.929–5.154) | 0.073 | 1.181 (0.384–3.635) | 0.772 |
| CD5 positivity | 2.446 (1.307–4.577) | 0.005* | 1.766 (0.821–3.798) | 0.145 |

DLBCL, diffuse large B-cell lymphoma; LDH, lactate dehydrogenase; OS, overall survival; OR, odds ratio; CI, confidence interval.

* P ≤0.05

**Table 3. Univariable and multivariable ananlysis for PFS in DLBCL.**

| Variables | Univariate analysis (PFS) | | Multivariate analysis (PFS) | |
|---|---|---|---|---|
| | OR (95% CI) | P-value | OR (95% CI) | P-value |
| *All DLBCL* | | | | |
| Age > 65yr | 25884 (1.738–3.854) | <0.001* | 1.918 (1.158–3.176) | 0.011* |
| Performance status | 3.288 (2.155–5.018) | <0.001* | 2.266 (1.383–3.713) | 0.001* |
| Elevated LDH | 2.363 (1.497–3.729) | <0.001* | 1.917 (1.100–3.341) | 0.022* |
| B symptom | 2.686 (1.766–4.085) | <0.001* | 2.191 (1.346–3.569) | 0.002* |
| > 1 Extranodal sites | 1.702 (1.079–2.685) | 0.022* | 0.534 (0.508–1.771) | 0.972 |
| Stage | 2.134 (1.424–3.199) | <0.001* | 1.192 (0.684–2.079) | 0.535 |
| Double-expressor | 1.722 (0.892–3.325) | 0.036* | 0.676 (0.264–1.732) | 0.414 |
| CD5 positivity | 1.748 (1.304–2.954) | 0.037* | 1.681 (0.873–3.239) | 0.121 |
| *Non-GCB DLBCL* | | | | |
| Age > 65yr | 3.186 (1.934–5.250) | <0.001* | 2.525 (1.402–4.548) | 0.002* |
| Performance status | 3.388 (2.003–5.731) | <0.001* | 1.570 (0.822–2.997) | 0.171 |
| Elevated LDH | 2.861 (1.613–5.075) | <0.001* | 1.752 (0.847–3.623) | 0.131 |
| B symptom | 2.665 (1.604–4.428) | <0.001* | 2.612 (1.443–4.728) | 0.002* |
| > 1 Extranodal sites | 1.949 (1.131–3.359) | 0.023* | 0.851 (0.365–1.983) | 0.708 |
| Stage | 2.598 (1.566–4.311) | <0.001* | 2.353 (1.264–4.380) | 0.007* |
| Double-expressor | 3.189 (1.500–6.782) | 0.003* | 1.297 (0.436–3.854) | 0.640 |
| CD5 positivity | 2.488 (1.347–4.598) | 0.004* | 2.840 (1.340–6.015) | 0.006* |

DLBCL, diffuse large B-cell lymphoma; LDH, lactate dehydrogenase; PFS, progression-free survival; OR, odds ratio; CI, confidence interval.

* P ≤0.05

found a strong association with CD5 positivity and BCL2 expression, which was reported in previous series [9, 13, 15, 23], and a correlation of CD5 with MYC, which has not been widely investigated before. However, only 2 (6.9%) cases of MYC rearrangement were detected and no MYC amplification or BCL2 rearrangement was identified in contrast to high levels of protein expression. Although there have been few reports, most revealed scarce mutations in the MYC gene in CD5+ DLBCL [9, 12]. Alteration of STAT3 and nuclear factor-kappaB (NF-kB) pathway was suggested for a possible explanation for overexpression of BCL2 in CD5+ DLBCL [9]. Similar genetic alterations have also been demonstrated in recent studies suggesting that the pathology of DE lymphomas reflect cumulative mutations involving B-cell receptor signaling and NF-kB [28, 30, 31]. In DE tumors, alternative transcriptional mechanisms, posttranscriptional, or posttranslational pathways involving miRNA, histone binding proteins, and ubiquitination were proposed as the basis of MYC activation [32–35]. Although direct evidence of CD5 expression influencing BCL2 or MYC activation cannot be presented in the present study, the aggressive biologic behavior of CD5+ DLBCL might be related to the higher frequency of DE in this group.

CD5 is a glycoprotein mainly expressed on the membrane of mature T cells and is only dimly expressed on a subset of late stage hematogones/normal B-lineage precursors [36–38]. Most CD5+ DLBCLs may originate from early B-cells prior to the germinal center stage with some exceptions [39]. Recently published paper by a Japanese group revealed a significantly lower prevalence of *MYD88* and *CD79B* gene mutations in CD5+ DLBCL in contrast to other extranodal DLBCLs [40]. These findings support the idea that the cellular origin of CD5 + DLBCL might be different from that of ordinary DLBCL. Although the cellular origin is unclear, expression of CD5 is known to induce interleukin-10 production and has anti-apoptotic function maintaining B-cell survival [41], which is attributed to the aggressive biology of CD5+ DLBCL. Currently, there is no specific guideline on management of CD5+ DLBCL, although many cases are controlled under high risk stratification. Based on recent studies suggesting that MYC upregulates immune checkpoint pathways such as programmed death-ligand 1 (PD-L1) [42], immunotherapy can be added to CD5+ DLBCL patients overexpressing MYC.

In summary, CD5+DLBCL is a distinct subgroup of DLBCL, characterized by aggressive clinical courses that can easily be missed in routine clinical practice. To date, this is the largest study of CD5+ DLBDLs in Korea, and we confirmed that CD5 is an easily available, cost-effective prognostic biomarker for DLBCL. Although the exact mechanism of CD5 expression and its connection to BCL2 or MYC and DE requires further investigation at the molecular level, our results will provide a basis for new therapeutic regimens including small molecule inhibitors that block MYC or BCL2 and even immunotherapy.

## Supporting information

**S1 Table. Clinicopathologic dadta of the CD5+ and CD5- DLBCL patients included in the analysis.**
(XLSX)

## Author Contributions

**Conceptualization:** Hee Young Na, Ji Eun Kim.

**Data curation:** Hee Young Na, Ji-Young Choe, Sun Ah Shin, Ji Eun Kim.

**Formal analysis:** Hee Young Na, So Hee Oh, Ji Eun Kim.

**Funding acquisition:** Hyun-Jung Kim, Jae Ho Han, Hee Kyung Kim, Ji Eun Kim.

**Investigation:** Hee Young Na, Ji-Young Choe, Sun Ah Shin, Hyun-Jung Kim, Jae Ho Han, Hee Kyung Kim, Ji Eun Kim.

**Methodology:** Ji-Young Choe, Sun Ah Shin, Hyun-Jung Kim, Jae Ho Han, Hee Kyung Kim, So Hee Oh.

**Supervision:** Ji Eun Kim.

**Validation:** So Hee Oh.

**Writing – original draft:** Hee Young Na, Ji Eun Kim.

**Writing – review & editing:** Hee Young Na, Ji-Young Choe, Sun Ah Shin, Hyun-Jung Kim, Jae Ho Han, Hee Kyung Kim, So Hee Oh, Ji Eun Kim.

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
