## [Decision Letter · Decision Letter 0]

18 Sep 2019

PONE-D-19-18460

Characteristics of CD5-positive diffuse large B-cell lymphoma among Koreans: high incidence of BCL2 and MYC double-expressors

PLOS ONE

Dear Dr. Kim,

Thank you for submitting your manuscript to PLOS ONE. After careful consideration, we feel that it has merit but does not fully meet PLOS ONE’s publication criteria as it currently stands. Therefore, we invite you to submit a revised version of the manuscript that addresses the points raised during the review process.

This manuscript is now essentially acceptable, but it would be further improved if the authors respond to the remaining issues noted by the reviews.

We would appreciate receiving your revised manuscript by 10/17/2019. To enhance the reproducibility of your results, we recommend that if applicable you deposit your laboratory protocols in protocols.io, where a protocol can be assigned its own identifier (DOI) such that it can be cited independently in the future. For instructions see: http://journals.plos.org/plosone/s/submission-guidelines#loc-laboratory-protocols

We look forward to receiving your revised manuscript.

Kind regards,

Joseph S Pagano

Academic Editor

PLOS ONE

Journal Requirements:

2. We noticed you have some minor occurrence(s) of overlapping text with the following previous publication(s), which needs to be addressed:

https://dx.doi.org/10.1371%2Fjournal.pone.0215103

https://doi.org/10.1038/modpathol.2016.128

https://doi.org/10.1111/his.12760

In your revision ensure you cite all your sources (including your own works), and quote or rephrase any duplicated text outside the Methods section. Further consideration is dependent on these concerns being addressed.

Additional Editor Comments:

This manuscript is now essentially acceptable, but it would be further improved if the authors respond to the remaining issues noted by the reviewers.

Reviewers' comments:

Reviewer's Responses to Questions

**Comments to the Author**

1. Is the manuscript technically sound, and do the data support the conclusions?

Reviewer #1: Yes

Reviewer #2: Yes

2. Has the statistical analysis been performed appropriately and rigorously? 

Reviewer #1: Yes

Reviewer #2: Yes

3. Have the authors made all data underlying the findings in their manuscript fully available?

Reviewer #1: Yes

Reviewer #2: Yes

4. Is the manuscript presented in an intelligible fashion and written in standard English?

Reviewer #1: Yes

Reviewer #2: Yes

5. Review Comments to the Author

Reviewer #1: The authors detail the clinicopathologic characteristics of CD5-positive diffuse large B-cell lymphoma (DLBCL) in Korea. The manuscript is well-written and the data are well presented. The manuscript would benefit from some clarifying details, particularly to aspects of the Methods and Results section, as detailed below.

1. Consider adding the following terms to the Keywords if space permits: cell of origin, MYC, BCL2

2. Please clarify the statement in the Abstract and Introduction that “CD5+ DLBCL had been introduced as a unique immunophenotypic subgroup of DLBCL in the 2008 WHO classification of haematolymphoid neoplasms, however, the revised 2016 version has omitted the designation of the CD5+ subtype.” In reviewing the text of the 2008 WHO Classification of Tumours of Haematopoietic and Lymphoid Tissues (4th edition), CD5+ DLBCL is not listed as a separate variant, subgroup, subtype or entity in Table 10.14 or in the overall Table of Contents, similar to the 2016/17 WHO Classification (revised 4th edition).

3. The authors state that the diagnosis of DLBCL was confirmed by two experienced hematopathologists. Was this also true for evaluation of immunohistochemical stains requiring semiquantitative analysis, such as BCL2, BCL6, CD10, MUM1, MYC and Ki67? How were these assessed and, if by more than one evaluator, how were discrepancies resolved, particularly if they fell close to positive/negative cut-offs?

4. A cut-off of 30% was chosen for BCL2 with reference to the Hans et al, Blood 2004 paper (reference 18) detailing the Hans algorithm to determine cell of origin in DLBCL. However, since BCL2 does not constitute part of the Hans algorithm, it would be more appropriate to use a cut-off for BCL2 that is typically used in the assessment of double-expressor phenotype, such as 50%, as supported by Johnson et al, J Clin Oncol 2012, which the authors cite as a basis for the MYC cut-off of 40%.

5. What cut-offs for positivity/negativity were used for scoring FISH in determining rearrangement and amplification? This should be detailed in the Methods section.

6. The authors exclude 1 case of CD5+ DLBCL that was transformed from chronic lymphocytic leukemia (CLL). Were the 166 CD5-negative DLBCL cases also de novo (i.e. were cases transformed from underlying low-grade lymphoma also excluded in the comparison group)? If so, this should be clearly stated. If not, this should be considered since cases of transformed DLBCL may behave differently from de novo cases.

7. Among the clinicopathologic features that differed significantly between CD5+ and CD5-negative cases was bone marrow involvement. Is it possible to review these cases to see whether there were any differences in patterns of marrow involvement between CD5+ and CD5-negative cases? Given the rarity of de novo CD5+ DLBCL, its prognostic significance and the fact that CD5 expression may not be routinely assessed by immunohistochemistry, any information about pattern of marrow involvement that might be a clue to CD5 positivity would be of practical utility to pathologists in helping to make this diagnosis.

8. On page 11, lines 185-186, the data provided for double-expressor phenotype in CD5+ vs. CD5-negative cases (9/29 vs. 7/166) is incorrect; these values are for MYC single expression as shown in Table 1 on page 10. Please correct.

Reviewer #2: The authors described clinicopathologic characteristics of CD5+ DLBCL among Koreans which should be the largest cohort in that area. Although this immunohistochemical subtype has been removed from the newest WHO classification of hematolymphoid neoplasm, it is still worthy of consideration in the clinical practice. It is well designed retrospective study and the manuscript was relatively well written. However, the authors should answer to the following inquires before publication.

Question 1.

What is the authors' suggested reasons that the incidence of CNS relapse in this cohort is far lower in comparison with Japanese's?

Question 2.

Are there any possibilities that CD5+DLBCL in this study might be high grade B cell lymphoma, not otherwise specified (not double hit lymphoma)?

Question 3

How frequent is double expressors (DE) among DLBCL? It is better to compare the incidence of DE and CD5+DLBCL to discuss possible link between them.

6. PLOS authors have the option to publish the peer review history of their article (what does this mean?). If published, this will include your full peer review and any attached files.

Reviewer #1: No

Reviewer #2: Yes: Gyeongsin Park

---

## [Author Response · Author response to Decision Letter 0]

25 Sep 2019

Dear Dr. Joseph S Pagano:

Academic Editor

PLOS ONE,

 Thank you and the reviewers for taking time to check the manuscript and suggest considerate comments. We have made some corrections and clarifications in the manuscript after going over the reviewer’s comments. And we’ve re-attached the manuscript file without blue highlighting along with the “manuscript with track change” file. We hope that the manuscript suffices publication in your journal.

In the following paragraphs, we address specific points in detail:

Reviewer #1:

1. Consider adding the following terms to the Keywords if space permits: cell of origin, MYC, BCL2

Response: We agree with your idea and added the three keywords you suggested above,.

2. Please clarify the statement in the Abstract and Introduction that “CD5+ DLBCL had been introduced as a unique immunophenotypic subgroup of DLBCL in the 2008 WHO classification of haematolymphoid neoplasms, however, the revised 2016 version has omitted the designation of the CD5+ subtype.” In reviewing the text of the 2008 WHO Classification of Tumours of Haematopoietic and Lymphoid Tissues (4th edition), CD5+ DLBCL is not listed as a separate variant, subgroup, subtype or entity in Table 10.14 or in the overall Table of Contents, similar to the 2016/17 WHO Classification (revised 4th edition).

Response: We’ve checked 2008 WHO classification of haematolymphoid neoplasms again, and confirmed that CD5+ DLBCL was listed as a separate immunophenotypic subgroup in Table 10.14. Although it was not designated as a separate entity, CD5+ DLBCL was listed as immunophenotypic variant of DLBCL, NOS, due to the aggressive behavior reported mainly by Japanese pathologists. And accumulating data from other Asian and Western countries also support this. However, we admit that using the word “distinct” in the abstract section and “unique” in the introduction section may cause confusion to the authors, thus we modified the statement. The same table cited in the literature is shown below to reassure this statement.

 (Ponz et al. J Hematopathol 2009.2:83-87.)

3. The authors state that the diagnosis of DLBCL was confirmed by two experienced hematopathologists. Was this also true for evaluation of immunohistochemical stains requiring semiquantitative analysis, such as BCL2, BCL6, CD10, MUM1, MYC and Ki67? How were these assessed and, if by more than one evaluator, how were discrepancies resolved, particularly if they fell close to positive/negative cut-offs?

Response: Since this is a retrospective study, IHC slides were reevaluated by the same hematopathologists (HYN and JEK) consecutively, with discussion of discrepancies. All cases reached consensus including those with the value near the cut-offs. We added this detail in the method section.

4. A cut-off of 30% was chosen for BCL2 with reference to the Hans et al, Blood 2004 paper (reference 18) detailing the Hans algorithm to determine cell of origin in DLBCL. However, since BCL2 does not constitute part of the Hans algorithm, it would be more appropriate to use a cut-off for BCL2 that is typically used in the assessment of double-expressor phenotype, such as 50%, as supported by Johnson et al, J Clin Oncol 2012, which the authors cite as a basis for the MYC cut-off of 40%.

Response: We appreciate you for raising this important issue. We’ve re-revaluated according to the criteria suggested by Johnson et al., which used 50% as the cut-off for BCL2. In CD5+ DLBCLs, one case was revised to negative BCL2 expression. In CD5- DLBCLs, we found 3 cases revised to negative expression. Nevertheless, there was no change in the number of double expressors (DE) in both CD5+ and CD5- cases because all DE cases showed diffuse BCL2 expression far exceeding 50%.

5. What cut-offs for positivity/negativity were used for scoring FISH in determining rearrangement and amplification? This should be detailed in the Methods section.

Response: We scored at least 100 cells, and the case was considered positive if 20% or more of the nuclei exhibited rearrangements or extra copies. The similar cut off was applied in other previous studies (Quesada et al. Modern Pathology 2017;30:1688-1697., Huang et al. Diagnostic Pathology 2019;14:81.)

6. The authors exclude 1 case of CD5+ DLBCL that was transformed from chronic lymphocytic leukemia (CLL). Were the 166 CD5-negative DLBCL cases also de novo (i.e. were cases transformed from underlying low-grade lymphoma also excluded in the comparison group)? If so, this should be clearly stated. If not, this should be considered since cases of transformed DLBCL may behave differently from de novo cases.

Response: Yes. All 166 cases of CD5 (-) DLBCL had no histologic and clinical evidence of transformation from precedent low grade lymphoma. We designed this study to reduce the bias as best as we could by including only de novo CD5 (-) cases. The incidence of SLL/CLL or follicular lymphoma is low in Korea, occupying 1.3% and 7.1% of total lymphomas, respectively, according to the most recent data (Jung HR et al, Classification of malignant lymphoma subtypes in Korean patients; a report of 4th nationwide study, Journal of Hematopathology 2019). Actually, transformed DLBCLs are extremely rare in Korean population.

7. Among the clinicopathologic features that differed significantly between CD5+ and CD5-negative cases was bone marrow involvement. Is it possible to review these cases to see whether there were any differences in patterns of marrow involvement between CD5+ and CD5-negative cases? Given the rarity of de novo CD5+ DLBCL, its prognostic significance and the fact that CD5 expression may not be routinely assessed by immunohistochemistry, any information about pattern of marrow involvement that might be a clue to CD5 positivity would be of practical utility to pathologists in helping to make this diagnosis.

Response: In Korea, bone marrow specimens are processed in the clinical laboratory medicine clinic which is a different department in every hospital. For reviewing the bone marrow slides, we have to ask help to clinical laboratory medicine doctors in three different hospitals, which would be very hard and time consuming. It would be appreciated if you would understand our situation.

8. On page 11, lines 185-186, the data provided for double-expressor phenotype in CD5+ vs. CD5-negative cases (9/29 vs. 7/166) is incorrect; these values are for MYC single expression as shown in Table 1 on page 10. Please correct.

Response: We corrected the typographical error (9/29  8/29, 7/166  5/166). 

Reviewer #2: 

1. What is the authors' suggested reasons that the incidence of CNS relapse in this cohort is far lower in comparison with Japanese's?

Response: Japanese studies reported 13 to 14.5% CNS relapse rate (Yamaguchi et al. Haematologica 2008; 93:1195-1202., Miyazaki et al. Annals of Oncology 2011;22: 1601–1607.). Notably, only one case of CD5+ DLBCL showed CNS relapse in our series. Kong et al. (J Korean Med Sci 2004; 19: 815-9) have also described 5 cases of de novo CD5+ DLBCL in Korean patients, all of whom showed bone marrow involvement but the authors did not mention the presence of CNS relapse. CXCR4/CXCL12 axis or downregulated ECM and cell adhesion related genes have been suggested as the potential causes for frequent CNS relapse in CD5+ DLBCL (Xu-Monette et al. Oncotarget 2015;6 (8):5615-5633., Barretina et al. Ann Hematol 2003; 82:500-505.), and further molecular studies may help in unraveling this discrepancy between Japanese and Koeran CD5+ DLBCL. 

2. Are there any possibilities that CD5+DLBCL in this study might be high grade B cell lymphoma, not otherwise specified (not double hit lymphoma)?

Response: According to the revised 4th edition of WHO classification, high grade B-cell lymphoma is diagnosed when clinically aggressive mature B-cell lymphoma shows features intermediate between DLBCL and Burkitt lymphoma (BL) or appears blastoid morphology. This category is referred to B-cell lymphoma, unclassifiable, with features intermediate between DLBCL and BL category in 2008 WHO classification. By definition, MYC, BCL2 and/or BCL6 rearrangements should not be found, and cases which can be morphologically diagnosed as DLBCL are excluded from this category. In our series, the morphology of CD5+ DLBCL cases clearly falls on classic DLBCL category and did not harbor blastoid appearance. Therefore, CD5+ DLBCL in our study should stay in DLBCL category.

3. How frequent is double expressors (DE) among DLBCL? It is better to compare the incidence of DE and CD5+DLBCL to discuss possible link between them.

Response: In the present study, DE was found in 6.7% (13/195) of all DLBCL. In CD5+ DLBCL, 27.6% (8/29) were classified as DE, while 3% (5/166) were categorized as DE in CD5- DLBCL. Although the overall incidence of DE in our cohort is lower than previous studies (Johnson et al. J Clin Oncol 2012, Bogusz et al. 23% PLoS One. 2017; 12(2): e0172364), the association of CD5 expression and DE seems reliable. As we mentioned in our manuscript, altered STAT3 and NF-kB pathway is assumed to be an explanation for overexpression of BCL2 in CD5+ DLBCL. This is so far the first series that shed light of the association of MYC and CD5 expression, and further studies are required to uncover the mechanism of MYC activation in CD5+ DLBCL. 

Editor’s comments:

Response: We corrected minor occurrences of overlapping text (paraphrase with citation of references) according to the editor’s advice. We submitted raw data of this study as a supplementary file. And we cited a new reference regarding the incidence of SLL/CLL and mantle cell lymphoma in Korea because it is so far the most recent data.

---

## [Decision Letter · Decision Letter 1]

9 Oct 2019

Characteristics of CD5-positive diffuse large B-cell lymphoma among Koreans: high incidence of BCL2 and MYC double-expressors

PONE-D-19-18460R1

Dear Dr. Kim,

We are pleased to inform you that your manuscript has been judged scientifically suitable for publication and will be formally accepted for publication once it complies with all outstanding technical requirements.

With kind regards,

Joseph S Pagano

Academic Editor

PLOS ONE

Additional Editor Comments (optional):

Reviewers' comments:

Reviewer's Responses to Questions

**Comments to the Author**

1. If the authors have adequately addressed your comments raised in a previous round of review and you feel that this manuscript is now acceptable for publication, you may indicate that here to bypass the “Comments to the Author” section, enter your conflict of interest statement in the “Confidential to Editor” section, and submit your "Accept" recommendation.

Reviewer #1: All comments have been addressed

Reviewer #2: All comments have been addressed

2. Is the manuscript technically sound, and do the data support the conclusions?

Reviewer #1: Yes

Reviewer #2: Yes

3. Has the statistical analysis been performed appropriately and rigorously? 

Reviewer #1: Yes

Reviewer #2: Yes

4. Have the authors made all data underlying the findings in their manuscript fully available?

Reviewer #1: Yes

Reviewer #2: Yes

5. Is the manuscript presented in an intelligible fashion and written in standard English?

Reviewer #1: Yes

Reviewer #2: Yes

6. Review Comments to the Author

Reviewer #1: (No Response)

Reviewer #2: Thank you, authors, for carefull responses with adequate correction to review comments.

I remind tnat the authors are supposed to keep ethics on research and publications.

7. PLOS authors have the option to publish the peer review history of their article (what does this mean?). If published, this will include your full peer review and any attached files.

Reviewer #1: No

Reviewer #2: Yes: Gyeongsin Park

---

## [Editor Report · Acceptance letter]

15 Oct 2019

PONE-D-19-18460R1 

Characteristics of CD5-positive diffuse large B-cell lymphoma among Koreans: high incidence of BCL2 and MYC double-expressors 

Dear Dr. Kim:

I am pleased to inform you that your manuscript has been deemed suitable for publication in PLOS ONE. Congratulations! Your manuscript is now with our production department. 

With kind regards,

on behalf of

Dr. Joseph S Pagano 

Academic Editor

PLOS ONE